# REAL-TIME LEARNING OF DECAY TRAJECTORY OF HIGGS BOSON USING RESERVOIR-IN-RESERVOIR ARCHITECTURE

## ABSTRACT

Real-time learning of the decay trajectory in Higgs bosons as they interact in the Higgs Field is the key to understanding and furthering of the *mass providing mechanism* and *particle interaction mechanism* beyond the Standard model in particle physics. We propose a novel machine learning architecture called *reservoir-in-reservoir* (R-i-R), to learn this complex high dimensional weak and electromagnetic interaction model involving a large number of arbitrary parameters whose full understanding remains elusive to physicists, making it harder to handcraft features or represent in a closed-form equation. Reservoir-in-reservoir is a reservoir computing (RC) approach, where we built a large reservoir using a pool of small reservoirs that are individually specialized to learn patterns from discrete time samples of decay trajectory without any prior knowledge. Each small reservoir consists of a paired primary and secondary reservoir of recurrently-connected neurons, known as learner and generator, respectively, with a readout connected to the head. During the training phase, we activate the learner-generator pairs within the pool. Then we excite each learners with an unit impulse and individual time windows of the incoming system. We train the internal recurrent connections and readouts using a recursive least squares-based First-Order and Reduced Control Error (FORCE) algorithm. To enhance adaptability and performance, we implement a time-varying forgetting factor optimization during training. This optimization helps control the fading and adaptation of the covariance matrix based on variations in the incoming decay trajectory and patterns. This comprehensive training strategy aims to guarantee that the entire reservoir pool evolves in harmony with the desired output dynamics. We optimize hyper-parameters such as the number of learner-generator pairs within the pool, their network sizes, batch sizes, and the number of training trials. During testing, we excite the generators in the pool, with only an unit impulse, to mimic the dynamic system. We facilitate real-time learning by re-triggering the training process involving learner-generator pairs whenever the error rate exceeds a predefined threshold. We evaluate our reservoir-in-reservoir architecture using Higgs boson decay trajectories as detected in the Compact Muon Solenoid (CMS) detector of CERN's Large Hadron Collider (LHC). The reservoir pool is used to model the dynamics of momentum components (and transverse momentum) as Higgs boson decays into photons and leptons (electrons and muons) with invariant masses between 120-130 GeV. Our results indicate that reservoir-in-reservoir architecture is a well suited machine learning paradigm in learning dynamical systems real time, with network size below state-of-the-art architectures and greater adaptability to aperiodic behavior.

## 1 INTRODUCTION

The study and emulation of systems dynamically evolving over space and time have been long-standing concerns in physics, engineering, and applied mathematics. Efficient control and replication of such nonlinear systems solely from observational data are essential endeavors. Machine learning (ML) systems have emerged to decipher intricate and adaptable structures. However, existing techniques for system identification typically rely on predefined model dynamics, frequently

leading to linear approximations that restrict their applicability to minor perturbations around equilibrium points within the dynamics limiting their effectiveness to small amplitude transient perturbations around a fixed point of the dynamics Nelles & Nelles (2020); Billings (2013). The discovery of the Higgs boson in 2012 Aad et al. (2012), crucial in particle physics, is one such space-time varying dynamic phenomenon, generated and studied through collisions rather than being found in isolation. Its subsequent decay into detectable particles are studied in detectors to understand the Higgs boson's interactions, seeking its role in the universe's formation, its mass generation mechanism, and its potential connections to dark matter and new particles Collaboration et al. (2012); Carpenter et al. (2014). But even the primary discovery of the particle required two and a half times more data than usual,to ensure Higgs boson had been discovered cms-publication-committee-chair@ cern. ch (2022). Reservoir Computing was proposed by Maass et al. (2002) as a human brain inspired computational paradigm to its preceding Turing Machines, with less resource constraints and easier training capabilities. They represent a pivotal advance in ML, addressing the complexities encountered in training traditional feed-forward networks. In contrast to layered architectures that necessitate intricate weight adjustments of every neuron across multiple layers, RC introduces a special form of Recurrent Neural Networks (RNN) with feedback mechanisms. While RNNs offer enhanced computational capabilities, their full training remains challenging due to back-propagation through time requiring entire sequence recalling, making it biologically inefficient as shown in Schmidt et al. (2019); Lillicrap & Santoro (2019); Hinton (2022). RC streamlines RNN training by introducing a fixed reservoir that requires no adjustment. This reservoir acts as parallel spatiotemporal filters applied to input signals, projecting nonlinear features into a high-dimensional space. The subsequent task of separating these features becomes a simplified linear process. Despite its apparent simplicity, RC-trained RNNs have demonstrated remarkable robustness across diverse applications, including data classification, systems control, time-series prediction, and the elucidation of linguistic and speech features. RC architectures are task-independent, utilizing property of high-dimensional dynamical systems (DS), statistical learning theory, and generic recurrent circuitry Maass et al. (2002); Seoane (2019); Lukoševičius & Jaeger (2009); Gauthier et al. (2021); Lukoševičius (2012).RC architectures Tanaka et al. (2019); Zhong et al. (2021); Moon et al. (2019); Abreu Araujo et al. (2020) have been explored to learn dynamically evolving systems, like temporal signals, electrical waves and more, for their adaptability to learn using less data and faster convergence. a) Echo state networks (ESN), exhibit the distinctive echo state property Jaeger (2002); Lukoševičius (2012), where solely the output weights undergo training, enabling rapid acquisition of temporal patterns. b)FORCE Architecture: The First Order Reduced Controlled Error based reservoirs are well-suited for temporal tasks capitalizing on the intrinsic spiking dynamics of neurons to acquire proficiency in processing sequential dataSussillo & Abbott (2009); Yada et al. (2021). c) Full-FORCE architecture extends the FORCE dynamics by amalgamating the spiking neuron dynamics with feedback connections, enhancing its aptitude for learning and controlling dynamical systems DePasquale et al. (2018). However, they face challenges of prior assumptions about the DS, large reservoir sizes, and slow convergence. They are stochastic, lack adaptability for aperiodic systems, and are primarily supervised. In response to these challenges, we present a real-time learning paradigm for system identification that eschews reliance on predefined equations. Our architecture, called *'reservoir-in-reservoir'* is a reservoir pool consisting of small learner a dynamically adjusts to evolving system dynamics, systematically reducing cost function through real time learning. This framework optimizes system attributes by considering input a-periodicity, concurrently keeping network sizes minimal to expedite convergence and enhance energy efficiency. Empirical validation underscores the competitiveness of our approach in learning aperiodic nonlinear systems. Our method demonstrates superior convergence capabilities with much reduced network dimensions.

## 1.1 The Higgs Boson Decay

The unification of two of the four fundamental forces which are the weak force and the electromagnetic force forms the basis of the Standard Model of particle physicsCowan (2012). It implies that electricity, magnetism, light and some types of radioactivity can be potential manifestations of a single underlying force known as the electroweak force. This unification of forces theory describes the electroweak force and its relationship with force-carrying particles, the photon, and the W and Z bosons McMahon (2008). Although photons emerge without a mass, W and Z have 100 times the mass of a proton. The Brout-Englert-Higgs mechanism gives a mass to the W and Z when they interact with an invisible field, called the "Higgs field", first proposed by Peter Higgs and later discovered in Aad et al. (2012). Following the Big Bang, as the universe cooled and its temperature

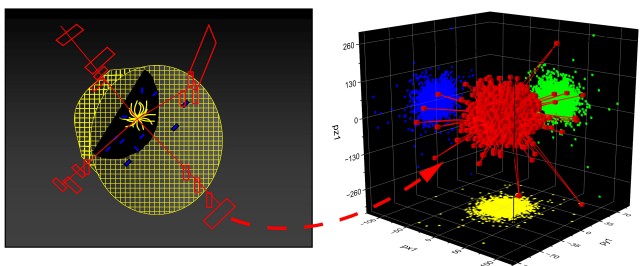

Figure 1: Left: An event display for the CMS experiment showing collision of particles occurring inside the detector. Right : the momentum trajectory of a single lepton in x,y,z plane visualized in 3 dimensions. The yellow, blue and green suggests the momentum variation on their respective planes of xy, yz and xz respectively.

dropped below a critical threshold, the Higgs field Bezrukov (2013) underwent continuous expansion, endowing particles with mass through their interactions. This process is mediated by the Higgs boson, which serves as the visible manifestation of the Higgs field and acts as the force carrier responsible for imparting mass to other fundamental particles.Extensive research that underwent by the ATLAS and CMS Collaborations at the Large Hadron Collider (LHC) Brüning et al. (2012) has been in characterising the properties of the Higgs boson, and unfolding all of the diverse ways in which this particle can decay. Being highly unstable, the particle decays into other subparticles and understanding this decay is of particular interest in the particle physics community. The most extensive but experimentally challenging is the Higgs decay to b-quarks : $H \rightarrow bb$ and the extremely rare decay is into four leptons (electrons or muons) :

$$H \rightarrow ZZ* \rightarrow 4l \tag{1}$$

Another rarest evidence is of the Higgs boson decaying to two leptons (either an electron or muon pair with opposite charge) and a photon. More information on the Higgs Boson decay equations are available in the Appendix. In our experiments we use equation 1 where the Higgs Boson decays into four leptons.

**Dataset Description :** For this study we used the Higgs candidate event database McCauley (2014) which provides a selection of Higgs candidate events. These events consist of an invariant mass falling within the range of 120-130 GeV, as made available by CMS. These events were selected and validated by the CMS Higgs Physics Analysis Group. The dataset comprises 10 gamma-gamma events, one 2e2mu event, one 4mu event, and one 4e event. Our dataset contained 3 Higgs candidate events (invariant mass between 120-130 GeV), where Higgs decays to four leptons. As input we provide the components of the momentum of the lepton (GeV) (px, py and pz variables) to the reservoir pool to learn and track the trajectory of a lepton Jomhari (2014).

**Related Work :** Higgs Boson decay using classical Machine learning (ML) has been studied in Jung et al. (2022); Cepeda et al. (2022) for probing exotic decay purpose. ML based methods have gained traction in processing data at particle colliders. With online filtering of streaming detector measurements,and offline analysis of data once it has been recorded Denby (1999); Sadowski et al. (2014). The ML Classifiers learn to distinguish between different types of collision events by training on simulated data from sophisticated Monte Carlo programs.Most of the existing state-of-the-art attempts have been to detect and classify the decay signals with respect to back ground signals using classical machine learning and quantum annealing Mott et al. (2017) techniques. In Sadowski et al. (2014) leverage the power of DNNs to provide the analyses of particle collider data, by learning high-level features from the data increasing the statistical power more than the common high-level features handcrafted by physicists. Non linear system identification using data driven methods is being studied rigorously since the discovery by Dzeroski & Todorovski (1995) for reproduction of underlying dynamics geometries and state space properties when there is absence of data by Brunton et al. (2016); Rudy et al. (2017); Sahoo et al. (2018); Sun et al. (2022). But sparsity explorations necessitates carefully defined candidate function library, prior system knowledge. Moreover, a linear combination of candidate functions may be insufficient for recovering complex mathematical expressions, especially aperiodic systems. As the library size increases, it empirically faces challenges in adhering to the sparsity constraint (Sun et al. (2022)). For Tree search methods in domains with high branching factors and deep search graphs, applying Monte Carlo Tree Search (MCTS) or any standard search algorithm becomes becomes challenging for real-time control, where an organized method to integrate knowledge becomes essential to limit the search to a manageable subtree

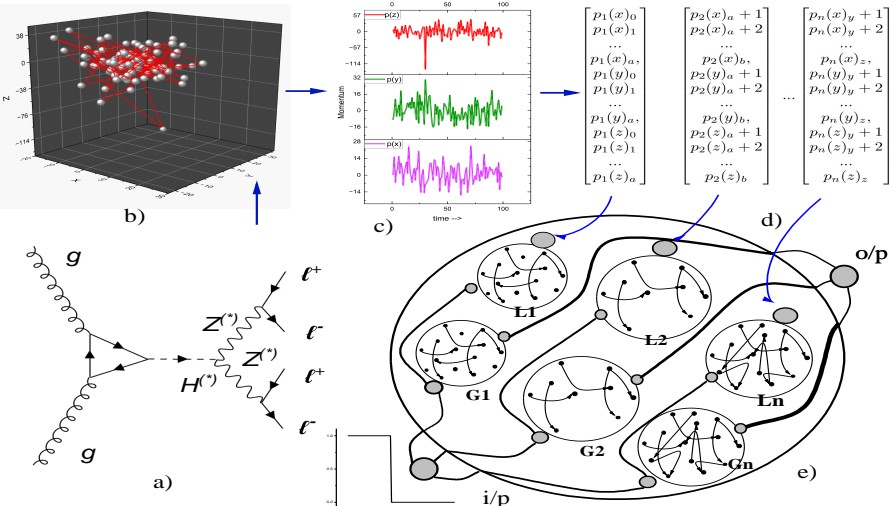

Figure 2: The Reservoir-in-Reservoir training phase : a)shows the decay from gluon to Higgs particle that decays further into 4 leptons.b)Each lepton decays further, whose momentum gets tracked in this dataset in x,y,z,direction in 3D space. c) The momentum p shown in x(red) y(green) and z (lavender) plane. d) The three individual matrices of momentum p(x), p(y), p(z) provided to three l-g pairs. e) The reservoir pool receives an input step impulse, the inputs from d) and trains the generators to produce the output from the readout, which is an union of the outputs from each l-g pair

(Browne et al. (2012)). Another complication arises when simulations demand significant CPU resources, and MCTS faces learning from a limited number of samples. Learning DS using RNNs have been proposed in Roweis & Ghahramani (2001); Lu et al. (2017); Duncker et al. (2019) based on inferring posterior over latent trajectories given a time sequence, but are harder to train, high memory intensive and exhibit minimal attractor tuning in case of aperiodically evolving DS. In the following sections, we describe the datasets used to learn the lepton momentum trajectory from Higgs Boson decay candidate events, provide a comprehensive architectural overview of a novel real-time learning and optimization approach in RC tailored to address the challenges posed by the modeling of unknown, data-driven aperiodic nonlinear systems. We follow it by discussing the results obtained and the observations we make with our rationale for the same.

## 2 METHODS

### 2.0.1 TRAINING THE RESERVOIR POOL :

We first introduce the initialization and training procedure of the reservoir pool, for which we must focus on the architecture of Figure 2. We conduct a search space exploration to find the optimal reservoir size inside the pool, the window size each learner-generator pair must process for optimal performance among other parameters using Algorithm 3, described in the Appendix section. Important to note here is that the pool can be initialized to any number of learner-generator pairs inside it, given the task we are opting to learn. In our case the optimal number of learner-generator pairs obtained by the architecture exploration were three. Each pair received three time windows each of length 2000 steps for training each time. Once the reservoir architecture search yields the system-specific optimal window length, reservoir size for the learner generator pairs, appropriate forgetting factor for specific time windows, the reservoir pool is initialized with an architecture similar to part e) of Figure 2. The activities of the pairs are given by 3. As shown in part e) of Figure 2 the reservoir pool is initialized with learner-generator (l-g) pairs of $L_i$, where $i = 1, 2, 3$ and 'n' in Figure 2 stands for 3 in our experiments (based on our architecture space exploration). The learners in each pair are initialized using equation 11 in ]1The reservoir pool is excited by a unit impulse. This unit impulse $u_{in}$ coupled with input weights $w_{in}$ excites each reservoir in each pair. It is important to note here that this is the only excitation that the generator reservoir receives, but that is not the case for the learner reservoir in the pair. The learner reservoir obtains a secondary input , which is the time

window of the system we are trying to learn. In our case this is the x-y-z trajectory of the lepton momentum (part d in 2). Instead of a single value each time step, the learner receives three values p(x), p(y) and p(z) each timestep. This delivers the ground truth into the pool through a vector of weights $w_{out}$. The components of both input weights are chosen from a uniform distribution between -1 and 1. Unlike state-of-the-art reservoir architectures where the ground truth is provided in the form of closed form equations (Sussillo & Abbott (2009); DePasquale et al. (2018); Lu et al. (2017); Lukoševičius et al. (2012)), in our case this is completely data driven, with no prior knowledge of the system dynamics. Hence this becomes a black box mimicking task or system identification by the reservoir pool, a much less explored area in RC. For the incoming aperiodic momentum trajectory of the lepton, our task is to train the reservoir pool to learn the changes in pattern in real time as shown in Figure 3b) and then mimic the system behavior with a single input impulse. Eventually the generator aims to merge its recurrent activity dynamics with signals representing the ground truth, which can be subsequently extracted by a linear readout in the form of :

$$p'(t) = w_{readout}^T(t)tanh(X_{Gn})(t) \qquad (2)$$

where $p'(t)$ denotes the output momentum at time $t$. The generator convergence : The neurons inside a reservoir exhibits the chaotic activity given by 15 :

$$\tau \frac{dX_G(t)}{dt} = -X_G(t) + C_G(tanh(X_G)) + w_{in}u_{in}(t), \qquad (3)$$

where $C_G$ is the N-unit connectivity matrix representing the sparse connections inside the reservoir. It's output is given by :

$$Z(t) = w_{readout}^T(tanh(X_G)(t)) \qquad (4)$$

Given our objective is to generate activity such that $z(t) \equiv p(t)$ , at time $t$ during training, before weight update, the error is given by :

$$e_-(t) = C_G(t-\Delta t)(tanh(X_{Gn}) - C_L(tanh(X_L)) - w_{out}p(t), \qquad (5)$$

Post weight update this error becomes :

$$e_+(t) = C_G(t)(tanh(X_{Gn}) - C_L(tanh(X_L)) - w_{out}p(t), \qquad (6)$$

As $t->\infty$, ideally $e_+(t)/e_-(t) -> 1$ at the end of training. And this is achieved by updating the connectivity matrix inside the generator reservoir by the delta rule Stone et al. (1986):

$$C_G(t) = C_G(t-\Delta t) - e(t)P(t)(tanh(X_L)), \qquad (7)$$

where the $P$ provides multiple learning rates to the presynaptic activity firing rates ($tanh(X_L)$ in each weight update by the following equation :

$$P(t) = \Lambda^{-1}(P(t-\Delta t) - \frac{P(t-\Delta t)tanh(X_{Ln}(t)tanh(X_{Ln})^T(t)P(t-\Delta t)}{\Lambda + tanh(X_{Ln}(t)^T P(t-\Delta t)tanh(X_{Ln}(t)}). \qquad (8)$$

where

$$\Lambda = \begin{bmatrix} \lambda_1 & 0 & \cdots & 0 \\ 0 & \lambda_2 & \cdots & 0 \\ . & . & . & . \\ \vdots & \vdots & \ddots & \vdots \\ 0 & 0 & \cdots & \lambda_N \end{bmatrix} \qquad (9)$$

It is to be noted here that p(t) is our desired output momentum trajectory and P(t) ($NXN$ matrix) is the running estimate of the inverse of the correlation matrix of the reservoir network activity rates scaled with the diagonal matrix of the forgetting factors, which can be expressed as : $P = (_t r(t)r^T(t) + \Lambda I)^{-1}$ Sussillo & Abbott (2009) Hence 6 can be expressed as :

$$e_+(t) = e_-(t)(1 - (tanh(X_G)^T(t)P(t)tan(X_G)(t)) \qquad (10)$$

In order to achieve convergence, or end of training, $(1 - (tanh(X_G)^T(t)P(t)tan(X_G)(t)) \to 1$ For low aperiodicty of incoming system, the subtrahend variable above may undergo a temporal evolution, initially closely approximating 1 and gradually converging asymptotically to 0 and remains consistently positive throughout the learning process. This behavior signifies a systematic reduction in error magnitude facilitated by weight updates, aligning with the intended learning objective. Ultimately, the ratio $\lim \frac{e_+(t)}{e_-(t)}$ tends towards 1. $\Lambda$ is a critical factor in this process and requires meticulous adjustment based on the specific characteristics of the target function. For lower excitation a value closer to 1 enhances the performance of the estimator (Fortescue et al. (1981)). While in case of high aperiodicity such as this, higher frequency and rapid changes of

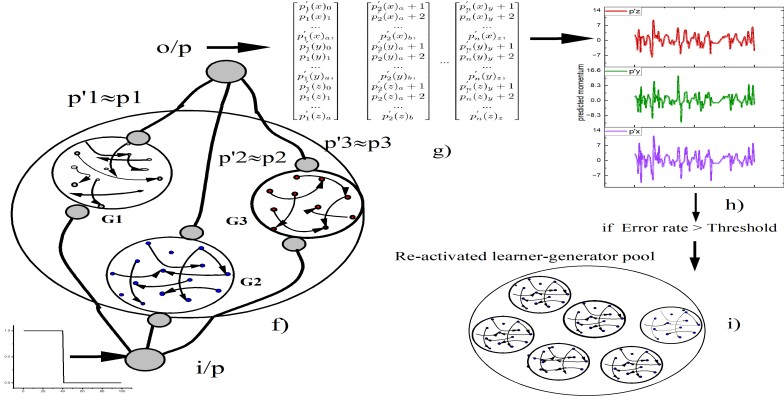

Figure 3: The Reservoir-in-Reservoir testing phase

Table 1: Model comparison: Mean squared error observed for different network sizes when comparing our architecture to state-of-the-art architectures

| Reservoir Size | ESN | FORCE | full-FORCE | Reservoir-in-Reservoir |
|---|---|---|---|---|
| 100 | 15.02 | 13.38 | 2.8 | 2.07 |
| 500 | 12.3 | 12.5 | 2.56 | 0.8 |
| 1000 | 10.8 | 14.98 | 2.54 | 1.98 |

target function need lower values to promote rapid learning but may introduce instability due to excessively swift weight adjustments. Hence, precise calibration of the forgetting factor is essential to achieve both stability and convergence in the learning processVahidi et al. (2005); Sussillo & Abbott (2009). High aperiodicity of target function increases the activity firing rate. Hence, the variable $(tanh(X_G)^T(t)P(t)tan(X_G)(t)$ in 10 can stray away from 0, leading to divergence from target function. Moreover, a very large reservoir size leads to higher instability in such scenarios because of high firing rate due to chaotic reservoir activity. This is where the efficacy of RiR lies in adaptability to incoming system dynamics, learning fewer timesteps at a time, adjusting the forgetting factor to facilitate efficient learning and re-triggering training for a learner-generator pair whenever the output and target diverge beyond threshold as shown in 2. This ensures a system specific data-driven approach to learning. Our objective lies in adjusting the recurrent connectivity matrix $C_G$ of each generator reservoir to internally generate signals equivalent to the ground truth time window provided to its respective learner in the pair (20, matching the mixing observed in its learner counterpart when exposed to an external p(t) input. We aim to align the combination of internal and external signals of the learner Li as $C_{Li} \, tanh(X_{Ln}(t) + w_{out}p_i(t)$, with the internally generated signal in the generator reservoir Gi , $C_{Gi}tanh(X_{Gn}$. This alignment is achieved by minimizing a designated cost function given in Equation 18 in 1. Fundamentally this minimization differs from the classical Recursive Least Squares (RLS) method in the approach to updating the covariance matrix P(t) (Equation 21). Within the classical RLS methodology, the covariance gradually converges to zero over time, leading to a loss in its capacity to adeptly capture alterations in parameters. Conversely, as depicted in Equation 21, the covariance matrix is subjected to division by a factor denoted as $\lambda$ (where $0.1 \leq \lambda \leq 1.0$) during each update. This deliberate division process effectively mitigates the rapid dissipation of the covariance matrix. And the value of $\lambda$ is not arbitrarily selected but chosen by the system during the reservoir architecture search mentioned earlier (3 in Appendix). The stepwise learning is provided in 1.

### 2.0.2 TRAJECTORY PATTERN GENERATION IN REAL TIME:

In the testing phase, the reservoir pool gets the unit impulse $u_{in}$ coupled with input weights $w_{in}$ as the only input trigger to start generating the momentum trajectory. As described in Algorithm 2, now the learner reservoirs are inactive and the generators are initialized with the recurrent connectivity matrix $C_G$ learnt using 20 during the training phase using RLS with forgetting. Given only an unit

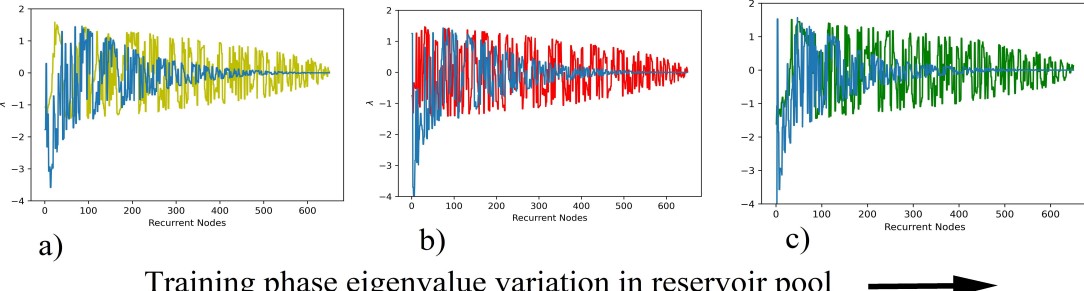

a)                                          b)                                          c)

Training phase eigenvalue variation in reservoir pool

Figure 4: The normalized eigenvalues before and after learning occurs. a) b) and c) representing generators 1,2 and 3 respectively. The blue lines show the stabilizing eigenvalues of the recurrent connectivity.y axis is normalized eigen value and x axis is recurrent nodes.

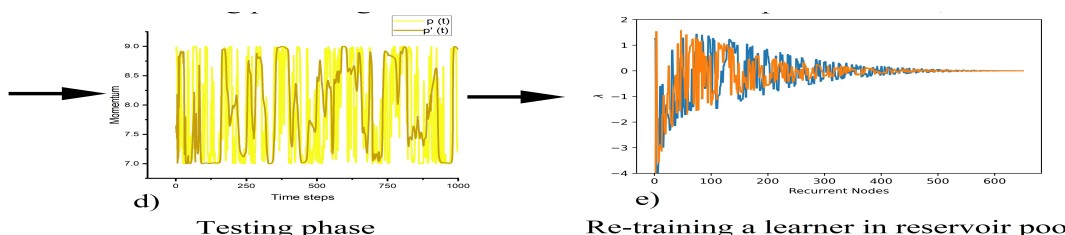

d)                                                          e)

Testing phase                              Re-training a learner in reservoir pool

Figure 5: d) showing the trajectory prediction in testing phase post which in e) re-training is triggered re initializing a learner generator pair but with pre-learned weights, making it stabilize faster than before

impulse, each generator reservoir generates an output in the form given in Equation 2. Given a set of optimal number of trials determined during the reservoir architecture search the NE and MSE is monitored over time. The pool now gets tested on its ability to generate patterns in the time windows assigned to each of them. The whole output is the union of the individual outputs generated by the individual l-g pairs. In our case we defined our threshold $> 1$. Upon exceeding this threshold the pool initiates an l-g pair to learn the new incoming system following the same steps of 1, except that instead of all l-g pairs being activated, only a single is activated. This keeps the re-training computation simple and less resource heavy. The detailed process to re-trigger training when total error exceeds threshold is available in Algorithm 2.

**Reservoir-pair Architecture search and optimization :** In our neural architecture search, we conducted experiments to find the optimal network size for learner-generator pairs in the reservoir pool, ranging from 100 to 1000 while maintaining a fixed window size. Our main objective was to balance network size and achieve state-of-the-art results, imposing a maximum network size constraint of 1000. Additionally, we explored the impact of forgetting factors (0.1 to 1.0), batch sizes (10 to 50), and the number of trials (10 to 50), enabling us to configure the reservoir pool with suitable network sizes and pairs tailored to specific patterns within a given window. This approach facilitated intelligent selection of the best-performing learner-generator pair during training, allowing real-time retraining, maintaining a compact network size, and enabling effective learning of previously unseen patterns without excessive computational costs or time delays.

**Model Comparison :** Our study evaluated the proposed architecture to SOTA reservoir architectures like ESN, FORCE, and full-FORCE. To ensure a fair comparison, we applied consistent evaluation criteria, testing all architectures with three sets of reservoir network sizes (100,500,1000 neurons) (1). Another reservoir size exploration (500, 650, and 1000 neurons) across 20 trials is presented in the Appendix (4). Training each model for 20,000 timesteps on momentum trajectory data for a single lepton, we then evaluated their performance on a separate unseen dataset of 10,000 timesteps of unseen trajectory data, while maintaining a forgetting factor of 0.1 in our R-i-R architecture to enhance learning dynamics and adaptability. We also experiment how varying forgetting factor affects the error rate for different reservoir sizes shown in 3 in Appendix. We have used the python 3 to program our learning architecture, the experiments have been run on Google Colab with a 51.0 GB System RAM, the model comparison to system identification algorithm baseline models have been made using the symbolic physics learner repository on Sun et al. (2023).

---

**Algorithm 1:** Training Learner-Generator reservoirs algorithm

---

**Input:** Incoming time series sequence from $t_0...t_T$ divided into n windows of $(1)p_1(t)$(from time $t_0...t_a), (2)p_2(t)$(from time $t_{a+1}...t_b), ...(n)p_n(t)$(from time $t_{y+1}...t_z$)

Initialize reservoir pool with n reservoirs $(L_1, L_2, ...L_n)$

**Requirement** :

Parameters = (Reservoir size=s,

Reservoir time step=dt, forgetting factor =$\lambda$, batch size =b,trials each batch=trials)

**for** *i=1,2,,, n* **do**

$\quad$ $L_i$ = Initialize each learner reservoir in the pool by the following equation

$$\tau\frac{dX_{Li}(t)}{dt} = -X_{Li} + C_{Li}(tanh(X_{Li})) + w_{in}u_{in}(t), \tag{11}$$

**end**

**Begin Parallel Training**

**Step 1** : Instantiate learners by providing input impulse and momentum signal with internal connections $C = C_L$

$$\tau\frac{dX_{L1}(t)}{dt} = -X_{L1}(t) + C_{L1}(tanh(X_{L1})) + w_{in}u_{in}(t) + w_{out}p_1(t), \tag{12}$$

$$\tau\frac{dX_{L2}(t)}{dt} = -X_{L2}(t) + C_{L2}(tanh(X_{L2})) + w_{in}u_{in}(t) + w_{out}p_2(t), ... \tag{13}$$

$$\tau\frac{dX_{Ln}(t)}{dt} = -X_{Ln}(t) + C_{Ln}(tanh(X_{Ln})) + w_{in}u_{in}(t) + w_{out}p_n(t), \tag{14}$$

**Step 2** : Instantiate generators in the pool by providing input impulse only with internal connections $C = C_G$

$$\tau\frac{dX_{G1}(t)}{dt} = -X_{G1}(t) + C_{G1}(tanh(X_{G1})) + w_{in}u_{in}(t), \tag{15}$$

$$\tau\frac{dX_{G2}(t)}{dt} = -X_{G2}(t) + C_{G2}(tanh(X_{G2})(t)) + w_{in}u_{in}(t), ... \tag{16}$$

$$\tau\frac{dX_{Gn}(t)}{dt} = -X_{Gn}(t) + C_{Gn}(tanh(X_{Gn})) + w_{in}u_{in}(t), \tag{17}$$

**Step 3** : Minimize Cost Function with forgetting factor for each of the above generator reservoir using following equations:

$$V(t) = \frac{1}{2}\sum_{i=1}^{t}\lambda^{t-i}((C_{Gn}(tanh(X_{Gn})) - C_{Ln}(tanh(X_{Ln}))) - p_t)^2, \tag{18}$$

$$e(t) = C_{Gn}(t - \Delta t)(tanh(X_{Gn}) - C_{Ln}(tanh(X_{Ln})) - w_{out}p_n(t), \tag{19}$$

$$C_{Gn}(t) = C_{Gn}(t - \Delta t) - e(t)P(t)(tanh(X_{Ln})), \tag{20}$$

$$P(t) = \frac{1}{\lambda}(P(t - \Delta t) - \frac{P(t - \Delta t)tanh(X_{Ln}(t)tanh(X_{Ln})^T(t)P(t - \Delta t)}{\lambda + tanh(X_{Ln}(t)^T P(t - \Delta t)tanh(X_{Ln}(t)}) \tag{21}$$

---

## 3    RESULTS AND DISCUSSION

Table 1 shows the comparison between our proposed R-i-R architecture to existing reservoir architectures ESN, FORCE, full-FORCE. Our architecture outperforms existing reservoir architectures in MSE evaluated for all three reservoir sizes. While ESN, FORCE and full-FORCE tend to perform better only with larger network sizes, R-i-R is designed to process an incoming system in parts, shared by its consisting l-g pairs. Hence for a given window length and its aperiodicity, R-i-R finds a sweet spot to perform optimally with adaptable forgetting. Larger network sizes in turn increase firing rate activities mentioned in the Methods section. It is to be noted here that our architecture is designed with the objective to reduce computation cost, keep network size to the minimal and yet achieve SOTA results. Table 2 shows the comparison with some SOTA system identification algorithms namely pySindy, GPLearn and MCTS.While pySindy outperforms GPLearn and MCTS, our architecture outperforms all the three. This performance improvement as a data driven approach for system identification, can be partly attributed to its little dependence on prior knowledge of the dynamics of particle decay. Additionally, the re-trigerring of training the l-g pairs with adapatable forgetting depending on changing aperiodicity and firing rates of the recurrent connectivity help keep the overall error rate low. The values of MSE reported are af-

---

**Algorithm 2:** Generating dynamic system pattern real time algorithm

---

**Begin Testing**

**Step 4** : Generate outputs from each reservoir in the pool

$groundtruth = p(t...T), totalerror = 0, totaloutput = 0, error = 0, Variance = 0,$
$totalVariance = 0, Window = W_i = 0, Threshold = threshold, timelength = tl$

**for** *i=1,2,,, n* **do**

    **for** *trials=1,... trials* **do**

$$p_i'(t) = w_{readout}^T(tanh(X_{Gn})(t)) \tag{22}$$

$$error = error + (p_i'(t) - p_i(t))^T(p_i'(t) - p_i(t)) \tag{23}$$

$$Variance = Variance + p_i(t)^T p_i(t) \tag{24}$$

    **end**

    $NE_i = error/Variance$

    $W_i = W_i \cup p_i(t_0...tl)$

**end**

$totaloutput = W_i$

$totalerror = totalerror + (totaloutput - groundtruth)^T(totaloutput - groundtruth))$

$totalVariance = totalVariance + groundtruth^T groundtruth$

$NE = totalerror/Variance$

**if** $NE > Threshold$ **then**

    Load $L_k G_k$ Learner-Generator reservoir pair

    k is determined based on which generator $(G_k)$ obtains lowest NE in previous steps of testing

    call **Training**

**end**

---

ter 10 trials each. This goes on to establish that this architecture shows promise in adapative real time system identification with a much lower memory requirement than a library driven or tree search based approach. In Figure 4, the transformations in the recurrent connectivity matrix $(C_G)$ within the generator reservoirs are observed before and after training, with a focus on its original form $(C_L)$. Initially, the eigenvalues of $C_L$ are predominantly clustered within a wide region of -4 to 2. After training, they converge within a smaller region of -2 and 1, with fewer than 500 recurrent nodes. Learner reservoirs with real parts initially exceeding 1 tend to gravitate towards real parts closer to 0 during the learning process, consistent with the earlier mentioned stabilization attribute. In testing phase, no alterations in the internal connectivity of the generator reservoirs takes place and only upon exceeding an error threshold, l-g pair is reactivated. This results in alterations to the existing eigenvalues, as depicted in subplot e) in Figure 5, but the convergence occurs much earlier, with approximately 350 recurrent nodes, ensuring comprehensive real-time stability.

## 4 CONCLUSION

Our novel RC paradigm representing an ML driven particle physics system identification, to elucidate Higgs Boson decay phenomena reveals that conventional RLS-driven loss function minimization may have limitations in achieving real-time adaptability to space-time varying nonlinear dynamic systems. We introduce a vector forgetting-based covariance matrix update tailored to mitigate covariance wind-up. Our R-i-R architecture pioneers real-time adaptability in data driven black-box learning context, offering significant network size reduction and enhanced performance. Upon acceptance, we will open-source our code-base for reproducibility, facilitating further insights and exploration of complex decay trajectories and parameters beyond lepton momentum.

Table 2: Mean Squared Error obtained from benchmarking with state-of-the-art system identification algorithms. Each metric obtained by 10 trials

| Algorithm | MSE |
|---|---|
| Reservoir-in-Reservoir (Ours) | 0.87 |
| pySindy Brunton et al. (2016) | 71.99 |
| GPlearn Ferreira et al. (2019) | 72.014 |
| Monte Carlo Tree Search Sun et al. (2022) | 72.009 |

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

## A  Appendix

### A.1  Higgs particle decay

There are three principle ways the Higgs boson particle is known to decay to a lepton pair and a photon: the leptons can be produced via an intermediate Z boson :

$$H \to Z\gamma \to ll\gamma \tag{25}$$

or a virtual photon :

$$H \to \gamma^*\gamma \to ll\gamma \tag{26}$$

or the Higgs boson can decay to two leptons :

$$(H \to ll) \tag{27}$$

with one lepton radiating a final-state photon.

Reservoir architecture search and optimization

---

**Algorithm 3:** Optimization Algorithm during training

---

**Input:** Incoming time series sequence from $t_0...t_T$
**Requirement** :
Window Length= wl = [1000, 2000, 3000, 4000, 5000]
Network size =s
**for** *i=0,1,,, 4* **do**
    $n = (wl[i])$
    **for** $\lambda$ *= 0.1,,,0.9* **do**
        **for** *s=100,,,1500* **do**
            Call Training loop in Algorithm 1
            error = Compute NE
        **end**
    **end**
**end**

---

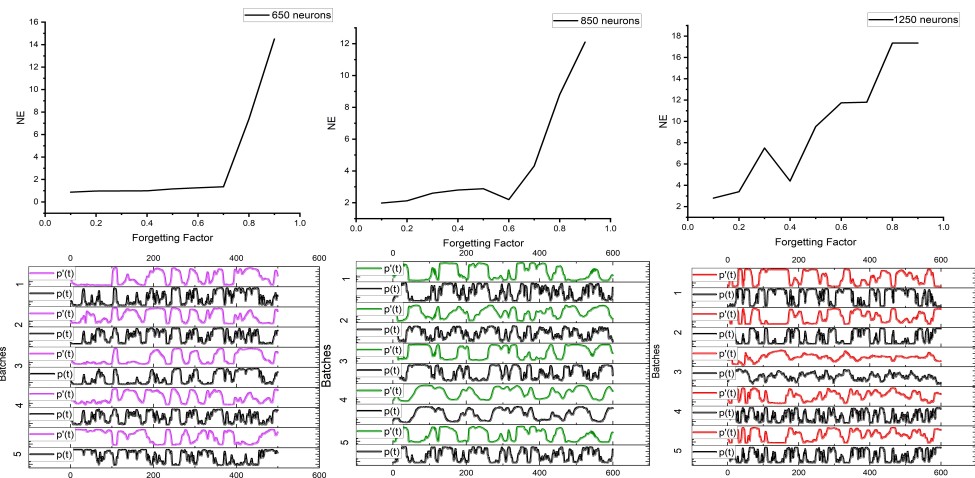

Figure 6: Our reservoir architecture search experiments. Row 1 (top) shows relationship between NE and forgetting factor for reservoir sizes 650,850,1250 (left to right). For a given set of unseen aperiodic incoming system, regardless of the recurrent connectivity network size of the l-g pairs we observe a jump in the normalized error rates as forgetting factor gets closer to 1. Row 2 (bottom) shows the momentum trajectory (a)-c)) of ground truth(black) vs predicted (colored) for 600 timesteps for each reservoir (a) reservoir 1, b) reservoir 2, c)reservoir 3. Each reservoir consists of 650 neurons in this row.

## A.2 EIGENVALUES OF THE RESERVOIR CONNECTIVITY BEFORE AND AFTER LEARNING :

**Eigenvalues of the reservoir connectivity before and after learning :**In the realm of eigenvalue analysis, we delved into the transformations that transpired within the recurrent connectivity matrix inside the generator reservoirs, $C_G$, before and after training, emphasizing on its original form, $C_L$. The eigenvalues of $C_L$ are initially observed to predominantly cluster within a wide region of -4 to 2 and post training the eigenvalues cluster within a smaller region of -2 and 1, converging with recurrent nodes as less as 500 as graphically depicted in Fig. 4. This is a particularly intriguing observation within each reservoir in the pool pertaining to eigenvalues. Each learner reservoir who originally possessed real parts much exceeding 1 exhibited a tendency to gravitate towards real parts closer to 0 during the learning process, a phenomenon consistent with the stabilization attribute earlier mentioned. During the testing phase there are no altercations made to the internal connectivity of the generator reservoirs, but upon exceeding error rate threshold (in our case 1.5 NE), a learner-generator pair gets reactivated to learn the new incoming pattern. This results in alteration of the already existing eigenvalues, but at a much smaller time as shown in subplot e) in Fig. 5. This time the convergence occurs much earlier, with 350 recurrent nodes approximately. This ensures comprehensive stability real-time. This gives reservoir-in-reservoir architecture the unique capability to harness the inherent expressive potential of a strongly connected pool of recurrent neural networks. Simultaneously, it performs the valuable function of pruning superfluous

| Reservoir Size | Trials | $\lambda$ | NE |
|---|---|---|---|
| 1250 | 10 | 0.1 | 2.8 |
|  | 20 |  | 2.6 |
|  | 50 |  | 2.05 |
| 1250 | 10 | 0.2 | 3.4 |
|  | 20 |  | 3.1 |
|  | 50 |  | 2.89 |
| 850 | 10 | 0.1 | 1.98 |
|  | 20 |  | 1.81 |
|  | 50 |  | 1.03 |
| 850 | 10 | 0.2 | 2.12 |
|  | 20 |  | 1.9 |
|  | 50 |  | 1.7 |
| **650** | 10 | 0.1 | **0.87** |
|  | 20 |  | **0.63** |
|  | 50 |  | **0.6** |
| **650** | 10 | 0.2 | **0.97** |
|  | 20 |  | **0.85** |
|  | 50 |  | **0.83** |
| 450 | 10 | 0.1 | 2.97 |
|  | 20 |  | 2.21 |
|  | 50 |  | 2.09 |
| 450 | 10 | 0.2 | 3.45 |
|  | 20 |  | 2.94 |
|  | 50 |  | 2.7 |

Table 3: **Error rates:** Normalized error rates of Proposed Reservoir-in-Reservoir architecture with forgetting factor $\lambda$ and reservoir size variation.

| Architecture | Reservoir Size | MSE |
|---|---|---|
| ESN | 1000 | 10.8 |
|  | 650 | 11.69 |
|  | 500 | 12.3 |
| FORCE | 1000 | 14.98 |
|  | 650 | 11.9 |
|  | 500 | 12.5 |
| full-FORCE | 1000 | 2.54 |
|  | 650 | 2.4 |
|  | 500 | 2.56 |
| **Reservoir-in Reservoir (Ours)** | 1000 | **1.98** |
|  | 650 | **0.75** |
|  | 500 | **0.8** |

Table 4: **Model comparison:** Mean squared error observed for different network sizes when comparing our architecture to state-of-the-art architectures.

Table 5: Training steps, reservoir pool size variation and effect on MSE

| Training steps | MSE |
|---|---|
| 5k | 2.6 |
| 10k | 0.9 |
| 20k | 0.87 |
| 50k | 0.87 |

| Number of l-g pairs in pool |  |
|---|---|
| 1 | 2.5 |
| 2 | 1.18 |
| 3 | 0.87 |
| 4 | 0.87 |
| 5 | 0.87 |

and potentially detrimental modes from the network's dynamic repertoire.

### A.3 ABLATION STUDY

We conducted an experiment to access how the reservoir pool performance varies over difference in length of training steps and difference in number of learner and generator pairs inside the pool shown in 5. While small amount of data (5k training steps) obtains a lower MSE, the system similar performance for 20k and 50k training steps.The ablation study based on amount of data has been tested on 3 l-g pairs with a forgetting factor of 0.2. Similarly we obtained same error rates beyond 3 l-g pairs with the lowest obtained for 1 pair in the pool only. Each of these cases in the number of pairs ablation study has been conducted with 20k training steps and forgetting factor of 0.2.

### A.4 RESERVOIR SIZE, FORGETTING FACTOR AND ERROR RATE, THE RELATIONSHIP :

Table 3 presents the key findings from our evaluation of the reservoir-in-reservoir architecture under varying conditions, encompassing reservoir size, trial counts, and forgetting factor selection. Our neural architecture search and optimization process guided us to a reservoir size of 650 neurons, with $\lambda = 0.1$ identified as the most effective choice for real-time estimation.Our observations highlight that in scenarios characterized by increased aperiodicity within time windows, higher trial numbers,

and the utilization of smaller forgetting factors, the architecture demonstrates superior performance, especially when deployed with a more compact network configuration. Notably, this contradicts the conventional approach seen in state-of-the-art reservoir architectures, which often advocate larger network sizes for tackling complex tasks DePasquale et al. (2018). Our reservoir-in-reservoir optimization learning framework demonstrates its ability to achieve $NE < 1.0$, even with a relatively modest reservoir size of 650 neurons and the adoption of forgetting factors set at 0.1 and 0.2.The RLS with forgetting mechanism has found application in parameter and system estimation across signal processing and engineering domainsPaleologu et al. (2008). Sub-optimal excitation can lead to covariance wind-up, causing loss of historical information, minimal integration of new data, and exponential growth of the covariance matrix, resulting in a sensitive estimator susceptible to numerical and computational errors. This issue becomes evident in scenarios involving the estimation of multiple parameters with varying rates of change, as in our mass and velocity estimation case, where a single forgetting algorithm is inadequate for tracking parameters with distinct variation rates. Therefore, the allocation of distinct forgetting factors to individual parameters becomes advantageous. Our vector-type forgetting or selective forgetting, introduced in Vahidi et al. (2005)is typically implemented in the covariance matrix update (refer to algorithm1 eqn. 21). Instead of uniformly dividing all elements by a single scalar $\lambda$, the covariance matrix P is adjusted by a diagonal matrix of forgetting factors. A critical departure from classical least squares methods relying on update mechanism for the covariance matrix P(t) which tends to zero over time, resulting in the loss of the ability to effectively track parameter changes. Our learning incorporates division by $\lambda < 1$ every update, mitigating the rapid attenuation of the covariance matrix. This adaptation enables exponential convergence, as substantiated in relevant academic literature.

Our findings suggest that the implementation of unique forgetting factors for each parameter offers a feasible strategy to address wind-up, allowing for effective real-time learning and adaptability in scenarios with varying levels of aperiodicity while maintaining a compact network size.

### A.5 DIFFERENCE BETWEEN EXISTING ARCHITECTURES AND RESERVOIR-IN-RESERVOIR :

ESNs offer more straightforward training algorithms when compared to other reserviir architectures. The training process primarily revolves only around updating the output readout weights in the following manner in addition to state update : $y(n) = f_{out}(w_{readout}(u(n), x(n), y(n-1)))$ where $u(n)$ is input units , $x(n)$ is hidden units ,and $y(n)$ is output units at time step n and $fout$ are the activation function of output units. The architecture avoids training recurrent connectivity and, while it mitigates the vanishing gradient issue in RNNs Hochreiter (1998) more effectively, it introduces instability and high variance due to randomly assigned connections within the reservoir, hindering the learning of complex, time-varying systems such as momentum trajectories in the xyz plane. In Table 4 we observe that ESN performs best for 1000 neuron reservoir size compared to FORCE and full-FORCE architecture which perform better for 650 neuron size. For learnic complex aperiodic trajectory, larger network sizes will be required to utilize ESNs. Our proposed architecture trains the learner-generator pairs during training using a specialized variation of the RLS based FORCE algorithm. The advantages of which are observed in Tables 3 and 4. Assuming perfect matching between learner and generator network activities and highly ideal generator outputs, akin to the original FORCE algorithm's proposed behavior, contrasts with practical scenarios where such perfect matches are rare. These discrepancies are most notable in principal component spaces, accounting for minimal recurrent activity variance, where even slight deviations between $tanh(x_L)$ and $tanh(x_G)$ substantially differentiate $C_L$ from $C_G$, affecting the stabilization of fluctuations. Both the original FORCE and full-FORCE architectures exhibit limited adaptability to real-time system changes, particularly aperiodic system learning, which constitutes their primary limitations. The limitations inherent in both the original FORCE and full-FORCE architectures primarily stem from their limited adaptability in the face of system changes real time, rendering them less suitable for aperiodic system learning. Reliance on the fundamental form of RLS for cost function minimization introduces significant performance reduction when dealing with dynamic environments evolving in both space and time, such as the momentum trajectory of the leptons.

