# OpenReview forum: "Real-time learning of decay trajectory of Higgs boson using reservoir-in-reservoir architecture"
_ICLR.cc/2024/Conference — Submitted to ICLR 2024_

### Official Review · Reviewer_f57r · 2023-10-30

**Soundness:** 2 fair
**Presentation:** 1 poor
**Contribution:** 2 fair
**Rating:** 1
**Confidence:** 4

**Summary:**

The paper proposes a modular reservoir computing approach, complemented by FORCE-based training, for learning the Higgs bosons decay dynamics.
The paper presents an empirical analysis in comparison to some other reservoir computing approaches.

**Strengths:**

- Potentially, the modular approach in reservoir computing seems interesting and holds promises in terms of effectiveness

**Weaknesses:**

- The paper's readability is really poor, due to several typos and incomplete descriptive content (just to mention one: figure 2 is not entirely described neither in the caption nor in the text).
- Similarly, the mathematical description of the proposed approach is rather vague and not given in detail through a proper section in the paper.
- Relevantly, it is not clear to me the motivations behind the proposal of the reservoir-in-reservoir in the given application context; why reservoir computing is even needed is not clear, given that other approaches for learning temporal dynamics are overall ignored.
- The experimental analysis seems not providing a proper model selection approach for the crucial hyper-parameters of both the proposed model and the methods from literature; moreover, std across repetitions is not reported.

**Questions:**

Several aspects of the proposed work remain unclear and not convincing, including the following that would need severe deepening:
- Thorough model selection on both the proposed model and the baselines from the literature;
- Comparison with fully trainable models outside the reservoir computing methodology;
- Motivation of the proposed architecture, and of the reservoir computing approach overall, to the tackled problem;
- Mathematical analysis of the introduced modular recurrent architecture;
- Profound restructuring of the manuscript, which has many typos and grammatical errors.

---

> ### Author Response · Authors · 2023-11-22
>
> Dear Reviewer,
>
> We thank you for your detailed review and questions of our work. We have tried to address your questions and weaknesses pointed out to the best of our abilities in the following.
> We have rewritten parts of the Introduction, Related works embedding into wider system identification literature. We have added descriptive caption to Figure 2. We have restructured the  Methods section, particularly included the learning convergence mathematics into the methods section to best explain our reservoir-in-reservoir training steps. This is in addition to the Algorithms that were available, to make the methods and algorithms tractable. We have added motivation into why this architecture has been chosen. We have also added Table 1 in the alignment that was suggested, more reservoir size exploration has been provided in the Appendix for the same. We also conducted a comparison of existing system identification algorithms pySindy, GPLearn and Monte Carlo Tree search which has been shown in Table 2. We have also updated our Results section to better describe our inferences from the tables and experiments section. We have tried to keep the sections precise and rewritten paragraphs (highlighted in blue) in the manuscript that has now been updated in the portal.
> We have also included an Ablation analysis of varying reservoir size, the training time steps and have reported the MSE in the Appendix section.
> Q&A Rebuttal :
> •	Thorough model selection on both the proposed model and the baselines from the literature;
> A : We have added Table 1 for comparison of our architecture to other existing reservoir architectures
> •	Comparison with fully trainable models outside the reservoir computing methodology;
> A : We have added Table 2 to compare our architecture to existing system identifiers outside RC methodology.
> •	Motivation of the proposed architecture, and of the reservoir computing approach overall, to the tackled problem;
> A : We have restructured and updated Introduction and methods section to address our motivation to this problem.
> •	Mathematical analysis of the introduced modular recurrent architecture;
> A : In addition to the Algorithm 1 and Algorithm 2 that we had submitted which contained the mathematical steps to the reservoir-in-reservoir learning, we have added a section (marked in blue) under Methods detailing the learning convergence achieved by our forgetting enabled RLS to minimize error.
> •	Profound restructuring of the manuscript, which has many typos and grammatical errors.
> A : We have restructured the Introduction, methods, Results and Discussion as well as Appendix. We have updated figure descriptions, addressed mathematical equations typos and grammatical errors to the best of our abilities.

---

### Official Review · Reviewer_vBzV · 2023-10-31

**Soundness:** 2 fair
**Presentation:** 2 fair
**Contribution:** 2 fair
**Rating:** 5
**Confidence:** 2

**Summary:**

The paper introduces a complex reservoir-computing based approach to to learning decay trajectories of Higgs bosons. It is argued that a reservoir computing approach is especially suitable in the real time setting needed to analyze experiments.

**Strengths:**

The approach is quite innovative and suitable for real time analysis of traces

**Weaknesses:**

The manuscript is not particularly accessible, I find, and the proposed learning mechanism is especially involved. Also, it is not clear why increasing the reservoir size results get worse.

**Questions:**

Why the algorithm is so sensitive to the choice of the reservoir size? The other approaches reported in Table 2 seem to depend significantly less on this choice

For a non specialist, it is very hard to assess the state of the art here, it would be helpful to understand what the reduced RMS found here imply in terms of classification of decays.

---

> ### Author Response · Authors · 2023-11-22
> **Response to Reviewer**
>
> Dear Reviewer,
>
> We thank you for your  review and questions of our work. We have tried to address your questions and weaknesses pointed out to the best of our abilities in the following.
> We have rewritten parts of the Introduction, Related works embedding into wider system identification literature. We have restructured the  Methods section, particularly included the learning convergence mathematics into the methods section to best explain our reservoir-in-reservoir training steps, its reasoning for sensitivity to the system being identified and how it affects the choice of  number of learner-generator pairs and their individual network sizes.. This is in addition to the Algorithms that were available, to make the methods and algorithms tractable. We have also added Table 1 in the alignment that was suggested, more reservoir size exploration has been provided in the Appendix for the same. We also conducted a comparison of existing system identification algorithms pySindy, GPLearn and Monte Carlo Tree search which has been shown in Table 2. We have also updated our Results section to better describe our inferences from the tables and experiments section. We have tried to keep the sections precise and rewritten paragraphs (highlighted in blue) in the manuscript that has now been updated in the portal.
> We have also included an Ablation analysis of varying reservoir size, the training time steps and have reported the MSE in the Appendix section.

---

### Official Review · Reviewer_Jf1c · 2023-11-02

**Soundness:** 2 fair
**Presentation:** 1 poor
**Contribution:** 2 fair
**Rating:** 5
**Confidence:** 3

**Summary:**

This paper proposes a Reservior-in-Reservior method to model the trajectories of the Higgs boson decay, especially in real time. Although the application is potentially an important one, the writting seems rushy and I believe the authors can do a better job at explaning the motivations for both the application and the method. The notations are a bit messy, so I'm not sure if I fully understand the method, please see the Questions part.

**Strengths:**

* Figures are illustrative, although can still be improved
* The background in physics is elaborated
* Contains necessary technical details to reproduce the work

**Weaknesses:**

* Notations are overall a bit messy. For example: (1) for math symbols, sometimes they are written as math, sometimes as regular texts. (2) In Figure 2(d), the subscript range from numbers (0,1) to letters ("a","y","z"). I can't spot a consistent pattern for the notation; (3) Figure 3 (g), subscripts are missing; (4) Eq. (2), a right parenthesis ")" is missing.
* Motivations are not unclear, see the Questions part.

**Questions:**

* What's the motivation for reservior-in-reservior? Is it similar to boosting methods, or ensembling methods in classical machine learning? Or perhaps similar to multi-head attention where each heads focus on some part of the problem, and then they are aggregated.
* It seems the goal here is to predict the trajectories of Higgs Boson decay (please correct me if I'm wrong). A more important problem is to detect Higgs Boson decay trajectory from other background trajectories. Can your method handle this?
* What's the motivation for having both a learner and a generator in the reservior? If I understand correctly, standard reservior only have one component instead of two. Is the idea similar to generative adversarial networks?

---

> ### Author Response · Authors · 2023-11-22
> **Response to Reviewer**
>
> Dear Reviewer,
> We thank you for your detailed review and questions of our work. We have tried to address your questions and weaknesses pointed out to the best of our abilities in the following. We have revised the text to remove any math symbols written as  regular text and inserted math symbols for the same. For the notation used in Figure 2d, the clarification is as follows :- Each l-g pair inside the pool receives a specific time window containing timesteps of the momentum p in the x, y and z direction. The first window consists of 0,1,2,3,…..a timesteps of p_{x}, p_{y}, and p_{z} respectively. The second window consists of a+1, a+2, a+3, a+4,…b timesteps of p_{x}, p_{y}, and p_{z} respectively. Likewise, for as many l-g pairs there will be in the pool, there will be as many windows received by the pool. In the figure the third matrix shows that for ‘n’  l-g pairs, the nth window must consist of y+1,y+2,y+3,y+4…z timesteps of p_{x}, p_{y}, and p_{z} respectively. And these momentums are annotated as p1(x), p2(x), p3(x)…pn(x) for all n windows for each direction of momentum.  We have updated Fig 3 to make the subscripts visible. We have put the right parenthesis that was earlier missing in Eq 2. We have restructured and updated our Introduction and Methods section to highlight and clarify our motivation behind this architecture and how it achieves learning convergence.
> Q&A
> A : R-i-R has been designed as a new paradigm of reservoir computing architectures to identify and mimic heavily aperiodic system transforming in a non linear space as a function of time, such as the Higgs particle decay. RC archs existing in literature are motivated to mimic the brain in terms of excitation through an external impulse and generating a desired output with that excitation, with little to no training of the internal sparse connectivity of the recurrently connected neurons inside. With the l-g pair in place within a pool, there is flexibility in breaking down an incoming data driven system into windows and provided into these pairs to learn the system by not just modifying the readout but also the internal connectivity if the reservoirs. The learning is facilitated by adaptive forgetting based on incoming aperiodicity if the system. Higher frequency of the incoming system generates more firing rate and fine tuning the forgetting in recursive lease squares  makes it more robust to noise, more adaptable to system change in real time and keep control on the loss. The training mechanism has been updated in the new Methods section with the algorithm, to clarify how the convergence is established. The key advantages of R-i-R are : 1. It is completely data driven, with no prior knowledge of the system being identified. 2. It is adaptable to system change with vector forgetting. 3. It achieves better performance than some data driven methods and existing RC archs based on the current application at hand. To answer how it is similar to other boosting methods or multi attention head, we must consider them as similar in the aspect that there is aggregation of individual outcomes at the last step to provide a desired output. They both can learn a target function in parts individually before the aggregation occurs. The key difference is in the prior knowledge required to model and understand sequential information. RC was proposed to mimic how the brain inspired computing works. Here, we take it up as a data driven system identifier where feature vectors cannot be assigned values and weights based on any prior knowledge. The motivation is to have a system identifier with minimal training, computationally inexpensive and robust to noise recurrently connected network of networks.
> A: We have attempted to simulate and identify the trajectory of the lepton momentum during Higgs particle decay phenomena. Since this has been dealt as a system identification and modelling problem, classification of trajectories was beyond the scope of this work. As a possible next step we plan to appl our architecture into diphoton trajectories of Higgs particles as well and plan to differentiate between the trajectories using our R-I-R. That problem will be dealt as a classification problem.
> A: The motivation for having pairs of learner-generator inside the pool stems from the problem that existing reservoir architectures have. Since only the readout layer is modifiable by training, it is hard to learn heavily aperiodic systems evolving over space and time, with little room for training internal connectivity. This problem was addressed using a similar mechanism that the full-FORCE mechanism addressed. By the introduction of a secondary network, we not only modify the readouts but the internal connection matrix with weight updates. The detailed description of the learning has been updated in the methods section and in Algorithm 1 and Algorithm 2 for stepwise understanding of the loss function and its minimization through forgetting recursive least squares.

---

### Official Review · Reviewer_Y31L · 2023-11-07

**Soundness:** 3 good
**Presentation:** 1 poor
**Contribution:** 2 fair
**Rating:** 6
**Confidence:** 4

**Summary:**

In the present work the authors present a new reservoir computing architecture centered around a larger reservoir with a number of smaller reservoirs embedded inside of it in an approach which can be likened to Multi-level, or coarse-graining approaches. This new algorithm is subsequently applied to system identification on Higgs boson decay trajectories from particle physics.

On this benchmark the new approach is compared to echo state networks, FORCE, and full-FORCE showing significantly better mean squared error than existing approaches.

**Strengths:**

Where the paper in its current form shines is the reflected view on the requirements necessitated by the particle physics identification problem at hand, the clarity of presentation of the used algorithms, and the incredibly well-crafted illustrations of the problem, and algorithmic components. The algorithmic approach itself is novel, and elicits the desire to explore its performance on other system identification problems.

**Weaknesses:**

The paper as is has a significant weakness in the clarity, and the style of its write-up. While the illustrations, as pointed out before, are incredibly well-done the flowing text lacks clarity, and focus. The introduction is very verbose, and _Related Work_ lacks embedding into the wider system identification literature. Where this shows the most is in the _Methods_ section, which lacks structure, and only is tractable due to the write-up of the algorithms.

Space invested in earlier sections, is then missing in the _Experiments_, and _Results_ section. The experiments section would benefit greatly from citations to the algorithms the approach is compared to, and could be more brief. The results section suffers from a number of issues in presentation such as
- The Eigenvalue testing being 2 pages ahead of the actual results section
- _Table 2_, the table with the main results being too small, and unpronounced. I would urge the authors to make it more prominent by e.g. rotating it by 90 degrees, and following the ICLR table formatting guidelines.
- The caption of Figure 6 is as is not useful, and requires a rewrite to make the results more clear, and better legible.

In addition the results section leaves a lot of questions unanswered with regards to the performance of the algorithm, further testing of which would improve the paper significantly:
- An ablation analysis over the number of reservoirs inside of the reservoir instead of just treating it as a hyper parameter, i.e. how does the quality of the result change over the number of reservoirs?
- Performance testing to substantiate the claim of _real-time learning_ by e.g. adding a third column to table 2 for the speed of the approach.
- Illustrative experiments complementing the presented theoretical analysis of recursive least squares-driven loss functions would help to further substantiate this contribution of the paper, and would strengthen the draft.

**Questions:**

- Why did you choose the 3 system identification approaches over other established approaches? Have you considered adding more comparisons to further system identification approaches?
- Why did you hide _"Notably this contradicts the conventional approach seen in state-of-the-art reservoir architectures, which often advocate larger network sizes for tackling complex tasks"_ in _Results and Discussion_ instead of making it much more prominent in the conclusion, the introduction, and possibly even the end of the abstract?
- What do you estimate the performance of your algorithm to be like in the low-data limit? Have you considered "starving" your algorithm (& others it is being compared to) of data to inspect their behavior?
- What tool did you use for the search space exploration for the optimal size of the pool? If it is a commonly used hyperparameter optimization tool, I would suggest to concretize this part, and cite the respective tool.
- Which framework, and based on which stack did you write your approach in? As is, this is left completely open, and would benefit from further clarification and citation to the respective libraries.

---

> ### Author Response · Authors · 2023-11-22
> **Response to Reviewer**
>
> Dear Reviewer,
>
> We thank you for your detailed review and questions of our work. We have tried to address your questions and weaknesses pointed out to the best of our abilities in the following.
> We have rewritten parts of the Introduction, Related works embedding into wider system identification literature. We have restructured the  Methods section, particularly included the learning convergence mathematics into the methods section to best explain our reservoir-in-reservoir training steps. This is in addition to the Algorithms that were available, to make the methods and algorithms tractable. We have also added Table 1 in the alignment that was suggested, more reservoir size exploration has been provided in the Appendix for the same. We also conducted a comparison of existing system identification algorithms pySindy, GPLearn and Monte Carlo Tree search which has been shown in Table 2. We have also updated our Results section to better describe our inferences from the tables and experiments section. We have tried to keep the sections precise and rewritten paragraphs (highlighted in blue) in the manuscript that has now been updated in the portal.
> We have also included an Ablation analysis of varying reservoir size, the training time steps and have reported the MSE in the Appendix section.
> Q&A Rebuttal :
> •	Why did you choose the 3 system identification approaches over other established approaches? Have you considered adding more comparisons to further system identification approaches?
>
> A : We have presently included three more system identification approaches to the comparison. While the earlier three were comparisons to SOTA reservoir architectures, additionally we have added non linear identification algorithms namely pySindy, GPLearn and Monte Carlo Tree search.
> •	Why did you hide "Notably this contradicts the conventional approach seen in state-of-the-art reservoir architectures, which often advocate larger network sizes for tackling complex tasks" in Results and Discussion instead of making it much more prominent in the conclusion, the introduction, and possibly even the end of the abstract?
>
> A: We have addressed this by adding the advantage of reservoir-in-reservoir architecture into our abstract, introduction and conclusion sections as suggested.
> •	What do you estimate the performance of your algorithm to be like in the low-data limit? Have you considered "starving" your algorithm (& others it is being compared to) of data to inspect their behavior?
>
> A: In the ablation study mentioned above we limit the training window to see how the testing performance varies  to different length of training steps of the system. We also test the performance on the number of l-g pairs in the pool.
> •	What tool did you use for the search space exploration for the optimal size of the pool? If it is a commonly used hyperparameter optimization tool, I would suggest to concretize this part, and cite the respective tool.
>
> A: We did not use any existing tool for the preparation of the optimal parameters in our architecture. Instead, we applied an algorithm provided as Algorithm 3 in the Appendix to measure
> •	Which framework, and based on which stack did you write your approach in? As is, this is left completely open, and would benefit from further clarification and citation to the respective libraries.
> A: We have highlighted our programming language framework with system specifics it was tested on in the methods section.

---

> ### Comment · Reviewer_Y31L · 2023-11-23
> **Thank you for addressing large parts of my concerns**
>
> I thank the authors for addressing a large number of concerns. In light of these improvements I have increased my rating.

---

### Comment · Area_Chair_1F1e · 2023-11-22

Dear all,

The author-reviewer discussion period is about to end.

@authors: If not done already, please respond to the comments or questions reviewers may further have. Remain short and to the point.

@reviewers: Please read the author's responses and ask any further questions you may have. To facilitate the decision by the end of the process, please also acknowledge that you have read the responses and indicate whether you want to update your evaluation.

You can update your evaluation positively (if you are satisfied with the responses) or negatively (if you are not satisfied with the responses or share other reviewers' concerns). Please note that major changes are a reason for rejection.

You can also keep your evaluation unchanged. In this case, please indicate that you have read the responses, that you do not have any further comments and that you keep your evaluation unchanged.

Best regards,
The AC

---

### Meta-Review · Area_Chair_1F1e · 2023-12-09

**Metareview:**

The reviewers have mixed opinions about the paper, but they overall tend towards rejection (6-5-5-1). The paper presents a new multi-level reservoir computing architecture to model the trajectories of Higgs boson decay. The reviewers have noted issues with the clarity of the presentation, suggesting that the paper could have been better written. The initial empirical evaluation was also limited, with claims that were not necessarily all supported by the results. Despite the lack of interactions from the reviewers, the authors have made a number of improvements to the paper during the author-reviewer discussion period. The authors have clarified the presentation, improved the empirical evaluation, and provided additional results. These changes, however, constitute a major revision of the paper, which would require a new round of reviews to be properly assessed. We encourage the authors to keep working on the paper and to submit a revised version to a future conference.

**Justification For Why Not Higher Score:**

Major revisions that would require a new round of reviews.

**Justification For Why Not Lower Score:**

N/A

---

### Decision · Program_Chairs · 2024-01-16

Reject